# Development and Characterization of Double-Antibody Sandwich ELISA for Detection of Zika Virus Infection

**DOI:** 10.3390/v10110634

**Published:** 2018-11-15

**Authors:** Liding Zhang, Xuewei Du, Congjie Chen, Zhixin Chen, Li Zhang, Qinqin Han, Xueshan Xia, Yuzhu Song, Jinyang Zhang

**Affiliations:** Faculty of Life Science and Technology, Kunming University of Science and Technology, 727 Jingming South Road, Kunming 650500, China; lidingzhang@aliyun.com (L.Z.); xueweidu@126.com (X.D); kmustccj@163.com (C.C.); arnoldchen1997@gmail.com (Z.C.); lilizhang7102@foxmail.com (L.Z.); qqhan10@kmust.edu.cn (Q.H.)

**Keywords:** NS1 protein, Zika virus, diagnosis, monoclonal antibodies, ELISA

## Abstract

Zika virus (ZIKV) is an emerging mosquito-transmitted flavivirus that can cause severe disease, including congenital birth defect and Guillain−Barré syndrome during pregnancy. Although, several molecular diagnostic methods have been developed to detect the ZIKV, these methods pose challenges as they cannot detect early viral infection. Furthermore, these methods require the extraction of RNA, which is easy to contaminate. Nonstructural protein 1 (NS1) is an important biomarker for early diagnosis of the virus, and the detection methods associated with the NS1 protein have recently been reported. The aim of this study was to develop a rapid and sensitive detection method for the detection of the ZIKV based on the NS1 protein. The sensitivity of this method is 120 ng mL^−1^ and it detected the ZIKV in the supernatant and lysates of Vero and BHK cells, as well as the sera of tree shrews infected with the ZIKV. Without the isolation of the virus and the extraction of the RNA, our method can be used as a primary screening test as opposed to other diagnosis methods that detect the ZIKV.

## 1. Introduction

The Zika virus (ZIKV), a new arbovirus, is a member of the *Flavivirus* genus of the Flaviviridae family. It was initially isolated from a macaque in 1947 in the Zika Forest of Uganda [1]. With fewer than 20 humans documented infected with the ZIKV, it received almost no attention before 2007. During this time, the ZIKV silently circulated in many parts of Africa and Asia without causing severe diseases or large outbreaks [2]. In 2015, an outbreak in Northeast Brazil led to an alarming number of babies born with microcephalus [3]. During this recent outbreak, many devastating severe diseases, including the Guillain−Barré syndrome in adults and congenital malformations in the fetuses of infected pregnant women such as microcephaly and fetal demise, were caused by the ZIKV [2,4,5]. Recently, the ZIKV has been recognized as a significant threat to global public health [6]. The disease was present in large parts of the Americas, the Caribbean, and also the Western Pacific region of Southern Asia during 2015 and 2016 [7,8]. Thereafter, the ZIKV spread rapidly and large-scale outbreaks were documented in other regions of the world [9]. As of April 2016, there were approximately 1.5 million people confirmed to be infected with the ZIKV. More than 46 countries have reported cases of ZIKV infections. In China, 13 ZIKV cases have been documented, and the possibility of new outbreaks still exists [10]. Mosquitoes of the *Aedes* species represent the main vector of transmission; however, it is possible to become infected with the ZIKV by exposure to blood, as well as perinatal and sexual contact [11,12]. Currently, there is no cure for ZIKV infection and no vaccine is available. Furthermore, rapid, efficient and easy-to-use kits are scarce [13]. Therefore, the early diagnosis of the ZIKV infection is the most effective way to treat patients and to control future outbreaks.

Presently, several studies have reported the methods used to detect the ZIKV. Using specific primers of viral RNAs for a highly-sensitive and simple experiment, the RT-qPCR assay was considered as a preferred diagnostic method. However, the false-negative results arising from new strains and the false-positive results arising from sample contamination still exist [14]. Therefore, other methods are needed to verify the accuracy of the RT-qPCR assay. Furthermore, there are other serological methods for detecting either ZIKV antigens (e.g., NS1) or immunoglobulins (e.g., IgG and IgM antibodies (Abs)). Due to the fact that IgM/IgG Abs, which are produced approximately seven days after the onset of symptoms, vary from patient to patient [15,16]. Thus, these methods are not suitable for the early diagnosis of ZIKV infection.

Nonstructural protein 1 (NS1) is an important protein secreted by cells infected with the virus, and it interacts with the host. It forms the homologous dimers within cells and binds to the type of adipocyte membrane system that participates in viral replication [17]. Furthermore, NS1 is a soluble protein that is secreted, suggesting that the virus can escape the immune system to strengthen interactions with the host [18,19]. More importantly, as the main antigen, NS1 can induce the production of Abs, which is important in early diagnosis of viral markers [20].

Currently, the early detection of the ZIKV largely depends on the NS1 protein, as several studies have reported that its level remains elevated up to nine days for Dengue, which is more sensitive than the other ZIKV proteins [21,22,23,24]. The detection of the ZIKV antigen for the development of a diagnostic method has not yet been reported, so the development of a ZIKV detection kit based on a specific monoclonal antibody (mAb) is absolutely critical [16].

In this study, we developed a rapid and sensitive method to detect the ZIKV in the supernatants and lysates of Vero and BHK cells, as well as the sera of tree shrew. Due to the short window for the detection of the ZIKV, it is presently difficult to diagnose patients with traditional methods [25,26]. Thus, we developed a double-antibody sandwich ELISA (DAS-ELISA) to detect the ZIKV-NS1 protein in individuals newly infected with the ZIKV and to assist the other screening methods used for detection. This method can effectively improve the diagnosis accuracy.

## 2. Materials and Methods

### 2.1. Expression and Purification of the NS1 Protein

The nucleotide sequence of the ZIKV-NS1 protein (Accession number: MG674719.1) was used to design the primers for the amplification of NS1. Total RNA was extracted from the supernatant of ZIKV infected Vero cells using the TIANamp Virus DNA/RNA Kit (TIANGEN, Beijing, China) and PCR was carried out to amplify the *NS1* gene with the specific primers 5′- GGAATTCGGGATGTTGGGTGTTCAGT-3′ (forward; underlined, *Eco*R I site) and 5′-CCGCTCGAGTTACGCTGTCACCACAGACCT-3′ (reverse; underlined, *Xho* I site). The amplified *NS1* gene product was digested with *Eco*R I and *Xho* I, and ligated into the pET-32a (+) to generate the recombinant cloning plasmid pET-32a (+)-NS1. The recombinant NS1 protein was expressed in BL21 (DE3) cells. Nickel-nitrilotriacetic acid (Ni2+-NTA) agarose resin was used to purify the NS1 proteins from the insoluble fraction of induced *E. coli* cells.

### 2.2. Ethics Statement

The animal study was approved by the Kunming University of Science and Technology with permit number: KMUST2018-0053. All experimental procedures involving mice and rabbits were performed in accordance with the regulations prescribed by the Administration of Laboratory Animals.

### 2.3. Production of Monoclonal and Polyclonal Antibodies Against the ZIKV-NS1 Protein

The production of mAb and polyclonal antibody (pAb) against ZIKV-NS1 protein was performed as previously described by our laboratory [27,28]. After repeated immunizations in rabbits, the polyclonal antiserum against the ZIKV-NS1 protein was successfully prepared. After the last immunization, the spleen cells of mice were collected and fused with SP2/0 cells. The fused cells were cultured in RPMI 1640 containing 20% FBS (Gbico, Gaithersburg, MD, USA) and hypoxanthine-aminopterin-thymidine (Sigma, Ronkonkoma, NY, USA) for one to two weeks. The ELISA and the indirect immunofluorescent assay were used to determine the presence of ZK-NS1-specific Abs in the hybridoma supernatants. Immunoglobulin G (IgG) from nine mAbs and the pAb were purified using protein A sepharose (GE Healthcare, Chicago, IL, USA).

### 2.4. Detection of Recombinant ZIKV-NS1 Protein by the DAS-ELISA

To screen a pair of Abs for the development of the DAS-ELISA, nine mAbs were conjugated with HRP, respectively. Horse radish peroxidase (HRP) labeled Abs and the pAb R1 were prepared pairwise combinatorial testing for pre-experiment. The pAb R1 was diluted to a final concentration of 10 μg mL^−1^ with CBS buffer (CBS; NaHCO_3_ 17.84 mM, Na_2_CO_3_ 27.64 mM, pH 9.6) and the wells of 96 plate were coated with 100 μL coated for 2 h at 37 °C. The wells were blocked with 200 μL blocking buffer (5% skim milk in PBS-T) for 2 h at 37 °C. Thereafter, 2 μg recombinant ZIKV-NS1 protein was added to each well and incubated for 2 h at 37 °C. The wells were washed three times with PBS-T (PBS-T; KCl 2.7 mM, KH_2_PO_4_ 2 mM, NaCl 137 mM, Na_2_HPO_4_ 10 mM, 0.05% Tween−20, pH 7.4), and then 2A7, 4G6, 3F1, 1F4, 3G12, 2C4, 1G2, 2C9, and 1F12 (all conjugated with HRP) were diluted to a final concentration of 1.4 μg mL^−1^ with blocking buffer. Thereafter, 100 μL of each mAb was added into each well, and the plate was incubated for 1 h at 37 °C. The wells were washed three times with PBS-T, 100 μL of the soluble TMB substrate solution (TIANGEN, Beijing, China) was added into each well and incubated for 20 min at 37 °C. Lastly, 50 μL of stopping buffer (2 M H_2_SO_4_) was added to terminate the reaction, and the optical density of each well was then measured at 450 nm using a microplate reader (Bio-Rad, Hercules, CA, USA).

### 2.5. Titer and Binding Affinity of mAb 1F12 and pAb R1 Based on the ELISA

An indirect enzyme-linked immunosorbent assay (ELISA) was used to determine the titer of mAb 1F12 and pAb R1. The wells of a 96-well plate were coated with 1 μg of the recombinant NS1 protein, and incubated for 2 h at 37 °C. The wells were washed three times with PBS-T, and then blocked with 200 µL of blocking buffer for 2 h at 37 °C. Thereafter, 100 µL of mAb 1F12 or pAb R1 at different dilutions (from 1:100 to 1:409,600) were added into each well, and the plate was incubated for 2 h at 37 °C. The wells were washed as described above, and then goat anti-mouse IgG (H + L) HRP or goat anti-rabbit IgG (H + L) HRP (GenScript, Piscataway, NJ, USA) was diluted to a final concentration of 2 μg mL^−1^ with blocking buffer. Thereafter, 100 μL of the secondary Ab was added into each well. The blank control well was incubated with 5% skimmed milk. The optical density of each well was measured at 450 nm using a microplate reader. The binding affinity of mAb 1F12 and pAb R1 were determined by the ELISA as described above.

### 2.6. Western Blot Assay

To characterize the reactivity and specificity of mAb 1F12 and pAb R1, Vero cells infected and mock-infected with ZIKV were harvested, and the proteins were separated by SDS-PAGE and transferred to nitrocellulose (NC) membranes. The membranes were blocked with blocking buffer for 2 h at 37 °C. Subsequently, the membranes were probed with mAb 1F12 or pAb R1 diluted to a final concentration of 1.4 μg mL^−1^ with blocking buffer and incubated for 2 h at 37 °C. Thereafter, the membranes were incubated with HRP-conjugated secondary Abs (diluted as above) for 1 h at 37 °C. After each incubation step, the membranes were washed five times with PBS-T. Finally, the EasySee Western Blot Kit (TransGen, Beijing, China) was used to detect the immunoreactive proteins.

### 2.7. Indirect Immunofluorescent Assay

For the immunofluorescent assay, ZIKV infected and mock-infected Vero cells were cultured for 48 h and fixed with pre-chilled acetone-methanol (1/1) for 20 min at −20 °C. After fixation, the cells were blocked with blocking buffer for 1 h at 37 °C. Subsequently, the cells were incubated with mAbs 2A7, 4G6, 3F1, 1F4, 3G12, 2C4, 1G2, 2C9, 1F12, or pAb R1 diluted to a final concentration of 1.4 μg mL^−1^ with blocking buffer, and 100 μL of each primary Ab was added into each well. The plate was incubated for 2 h at 37 °C. Finally, fluorescein isothiocyanate (FITC)-conjugated goat anti-mouse immunoglobulin G (Abcam, Cambridge, UK) was diluted to a final concentration of 1 μg mL^−1^ with blocking buffer, and 100 μL of the secondary Ab was added into each well. The plate was incubated for 1 h at 37 °C. After each incubation step, the cells were washed five times with PBS-T. Lastly, 200 µL of PBS was added into each well for fluorescent detection at 488 nm using the Leica DMI3000B microscope (Buffalo Grove, IL, USA).

### 2.8. Establishment of the DAS-ELISA Based on pAb R1 and mAb 1F12 Probe

To establish the DAS-ELISA, 1 µg of pAb R1 was diluted in CBS buffer and incubated for 2 h at 37 °C. The wells of a 96-well plate were blocked with blocking buffer for 1 h at 37 °C. Thereafter, the ZIKV-NS1 protein was added into each well, and the plate was incubated for 2 h at 37 °C. The wells were washed three times with PBS-T. Thereafter, mAb 1F12-HRP was diluted as described above and 100 μL of the secondary Ab was added into each well, followed by incubation for 1 h at 37 °C. After washing, 100 µL of TMB was added into each well, and the plate was incubated for 15 min at 37 °C. Lastly, 50 µL of stopping buffer (2M H_2_SO_4_) was added to terminate the reaction. The optical density of each well was measured at 450 nm using a microplate reader. A yellow-colored reaction and a high absorbance reading were indicative of a positive reaction, whereas a clear-colored reaction and a low absorbance reading were indicative of a negative reaction.

### 2.9. Specificity and Sensitivity of the DAS-ELISA

The specificity and sensitivity for the ZIKV-NS1 protein were determined based on the calibration curve of the DAS-ELISA. Vero cells were infected with four similar flavivirus (DEN-1, DEN-2, DEN-3, and JEV). The natural NS1 protein in cell culture supernatant was used to determine the specificity of the DAS-ELISA. Four similar flavivirus and the ZIKV were added into each well precoated with pAb R1, and the plate was incubated for 2 h at 37 °C. Thereafter, mAb-HRP probes were added and recognized with the antigen. After washing the plate three times with PBS-T, 100 µL of TMB was added, and the plate was incubated at 37 °C. Negative and positive samples could be easily distinguished by the naked eye after 15 min. The optical density of each well was measured using a microplate reader. The sensitivity of the DAS-ELISA was established using the recombinant ZIKV-NS1 protein serially diluted in water, for the final concentrations of NS1 protein from 500 µg mL^−1^ to 0.122 µg mL^−1^. The following steps were the same as those previously described. 

### 2.10. Application of the DAS-ELISA for the Detection of ZIKV Infection in Cells and Tree Shrews

The test for the application of the DAS-ELISA, supernatants, and cell lysates from Vero and BHK cells, as well as sera from three tree shrew infected with ZIKV were used. In brief, Vero and BHK cells were infected with one multiplicity of infection (MOI) and cultured in the presence of 1% FBS. The supernatants and cell lysates of infected Vero and BHK cells were collected at different times (12, 24, 36, 48, 60, and 72 h) and used for the DAS-ELISA assay. On the other hand, tree shrews were subcutaneous inoculated with 1 × 10^6^ PFU of the ZIKV. Tree shrews’ sera were collected at different infection times (from 1 to 15 days) and used for the DAS-ELISA assay described above.

## 3. Results

### 3.1. Expression and Purification of the ZIKV-NS1 Protein

The *ZIKV-NS1* gene was amplified from the cDNA of infected cells and cloned into the pET-32a (+) expression vector. The pET-32a-NS1 plasmid was transformed into *E. coli* Rosetta (DE3) cells, and the transformation was confirmed by PCR (Figure 1A, lane 1). Thereafter, the recombinant NS1 protein was expressed after induction with 1.0 mM IPTG for 10 h at 20 °C (Figure 1B, lane 1). The expressed NS1 protein was purified using a Ni2+-NTA resin column (Figure 1B, lane 3).

### 3.2. Evaluation of the Rabbit Antiserum

The recombinant NS1 protein was used as an immunogen for immunization. After repeated immunizations in rabbits, the antiserum was produced. The reactivity and specificity of the rabbit antiserum were evaluated by immunofluorescent and Western blot assays. As shown in Figure 2A, the anti-ZIKV-NS1 serum (R1) reacted with the native NS1 protein in ZIKV infected cells, and the negative control was not. The Western blot results showed that the rabbit pAb R1 specifically recognized the recombinant NS1 protein, as well as the native NS1 protein in cells infected with the ZIKV (Figure 2C,D).

### 3.3. Characterization of the Ascites Against NS1 Protein

Ascites of the nine hybridomas were prepared and purified using protein A Sepharose. The reactivity and specificity of the nine ascites were determined using an immunofluorescent assay and an ELISA. All mAbs were further verified by the immunofluorescent assay. The results showed that all Abs recognized the native NS1 protein in cells infected with the ZIKV, and the negative control had no fluorescence (Figure 2A). Furthermore, the results of the ELISA assay showed that the nine mAbs reacted with the recombinant NS1 protein (Figure 2B).

### 3.4. Establishment of the DAS-ELISA

Several Ab pairs were screened to develop the DAS-ELISA. In brief, the mAbs were labeled with HRP probe and used for ELISA (Figure 3A). Thereinto, HRP-labeled mAbs (2A7, 4G6, 3F1, 1F4, 3G12, 2C4, 1G2, 2C9, and 1F12) and unlabeled pAb R1 were prepared in different combinations for the pre-experiment. The absorbance values indicated that R1 and 1F12 were more effective than the other combinations (Figure 3B). Besides, R1 and 1F12 were titrated by the ELISA (Figure 3C,D). Horse radish peroxidase (HRP) labeled mAb 1F12 and pAb R1 were used for subsequent specificity and affinity experiments. The K_D_ values of HRP-labeled mAb 1F12 and pAb R1 were measured using an affinity test, and the results were analyzed by non-linear regression using software GraphPad Prism 6. Their K_D_ values were calculated as K_D_ = 1.77 ± 0.206 nM for the HRP-labeled mAb 1F12 and K_D_ = 0.868 ± 0.0816 nM for pAb R1 (Figure 3E). Lastly, pAb R1 and mAb 1F12 were purified, and their purity was confirmed by SDS-PAGE (Figure 3F). According to Figure 3B and Figure 3E, pAb R1 functioned as the capture Ab and mAb 1F12 functioned as the signal Ab, and they were the best combination for detecting the ZIKV.

### 3.5. Optimization Operations

To achieve a high sensitivity, a short detection time, and a low cost, the individual steps of the DAS-ELISA were optimized. Firstly, a suitable buffer for pAb R1 was identified using three commonly employed coating buffers (PBS buffer, TBS buffer, and CBS buffer). As shown in Figure 4A, CBS buffer had the highest coating efficiency. Secondly, the time and temperature for the immobilization of pAb R1 onto the plate were optimized. Figure 4B showed that the absorbance was higher at 37 °C than at 4 °C with increasing incubation time from 60 to 120 min. However, there was no significant difference when the incubation time or temperature was raised. Thirdly, the optimal concentration of pAb R1 and mAb 1F12 was determined when R1 and HRP-1F12 Ab s were serially diluted and incubated with a fixed amount of the recombinant NS1 protein, respectively. The results showed that the capturing efficiency of the R1 and HRP-1F12 Ab s increased between 10 µg mL^−1^ and 15 µg mL^−1^. There was no obvious change in the curve at concentrations greater than 10 µg mL^−1^ and 15 µg mL^−1^, respectively (Figure 4C,D). Moreover, we also established the temperature and time of antigen binding for pAb R1. The results of the ELISA indicated that the antigen was able to be completely bound by R1 at 37 °C (Figure 4E). Lastly, the binding of the HRP-1F12 probe to the recombinant NS1 protein was investigated when pAb R1 was combined with the NS1 protein and the HRP-1F12 antibody were added into the wells, and incubated at 37 °C or 4 °C, for different time periods. The results showed that the suitable time and temperature were 1 h at 37 °C (Figure 4F).

### 3.6. The Specificity and Sensitivity of ZIKV Detection by DAS-ELISA

The specificity of the DAS-ELISA was evaluated with others similar flaviviruses. As shown in Figure 5A,B, the presence of the ZIKV and the recombinant ZIKV-NS1 protein were defined by the yellow color and the optical density of the solution, respectively. At the same time, the sensitivity of the DAS-ELISA was established using the recombinant NS1 protein serially diluted from 500 µg mL^−1^ to 0.122 µg mL^−1^. Figure 5C,D clearly show that the color of the solution and the optical density gradually reduced with decreasing concentrations of the recombinant NS1 protein. Figure 5E shows a good linear relationship when the concentration of the NS1 protein ranged from 500 µg mL^−1^ to 0.122 µg mL^−1^, with a correlation coefficient of 0.9984. Within this range, the DAS-ELISA is useful for quantitative analysis, with a sensitivity of 0.122 µg mL^−1^.

### 3.7. Application of the DAS-ELISA for ZIKV Detection in the Supernatants, Cell Lysates of Cells, and the Sera of Tree Shrews

To test the application of the DAS-ELISA, supernatants, and lysates from Vero and BHK cells, as well as sera of tree shrews infected with ZIKV were used. In brief, Vero and BHK cells were infected with viruses at one multiplicity of infection (MOI). The supernatants of the infected cells were collected at different infection times and concentrated. Thereafter, 100 µL of the supernatant or cell lysates was used for DAS-ELISA assay, respectively. The results of the DAS-ELISA showed that the supernatants and lysates of Vero and BHK cells collected at 12 h, 24 h, 36 h, 48 h, 60 h, and 72 h were detectable (Figure 6A–D). Furthermore, the ZIKV was used to infect tree shrews. Thereafter, tree shrews’ sera were collected at different infection times for DAS-ELISA. Figure 6E,F shows that the DAS-ELISA detected the ZIKV, even at eight days after infection. Moreover, the ZIKV-infected supernatants and sera were used for RT-qPCR using specific primers and probes (Table 1) to confirm the existence of ZIKV (Figure 6G,H), whereas the Western blot assay was carried out to detect the NS1 protein in the supernatants and lysates of cells (Figure 6I).

## 4. Discussion

The ZIKV is an emerging mosquito-transmitted flavivirus that causes severe disease, including congenital birth defects during pregnancy [3]. Recently, an outbreak of the ZIKV in Northeast Brazil led to an alarming number of babies born with microcephaly in this region [29]. Based on the current clinical manifestations, it is very difficult to detect the ZIKV, especially in the early stages of infection. Regardless, the early diagnosis of ZIKV infection is important because it is a major factor in the treatment of patients and in the control of future outbreaks [30].

Traditional diagnostic methods that involve serological techniques and virus isolation are considered as the golden methods. However, these methods are time consuming, with low specificity. Fluorescence quantitative polymerase chain reaction (PCR) has been widely used in the detection of the ZIKV, due to its high sensitivity and specificity [31]. Although this method is reliable, most regions of the world do not have access to the expensive equipment, as well as the stable power required to operate the system. Due to the limited window period of virus detection, it is difficult to diagnose patients one to two weeks after the onset of symptoms [25,32].

Thus, it is very important that this issue of virus detection be addressed. A rapid, accurate, and sensitive diagnostic technique is urgently needed for early detection of ZIKV-infected patients, which can effectively control ZIKV transmission.

In this study, the NS1 protein was purified and used as antigen to immunize rabbit and mice. Nine mAbs and one pAb were prepared and screened using ELISA, Western blot, and indirect immunofluorescent assay. Based on the two Abs, the DAS-ELISA was developed for the detection of the ZIKV-NS1 protein. The specificity and sensitivity of the DAS-ELISA were successfully demonstrated. Moreover, the practical application of the DAS-ELISA was also determined using the supernatants and lysates from Vero and BHK cells, as well as the serum of tree shrews that were infected with ZIKV. Moreover, all samples used for the DAS-ELISA assay were screened by RT-qPCR and Western blot to confirm the existence of the ZIKV. The results showed that ZIKV existed in the supernatants of infected Vero, BHK cells, and the sera of infected tree shrews. In addition, we observed the natural ZIKV-NS1 protein secreted into the cell culture supernatant. Furthermore, the ZIKV tree shrew model confirmed the accuracy of the DAS-ELISA we established. Figure 6E,F revealed that the native ZIKV-NS1 protein was present in the serum from two to eight days after infection. However, we found that the RNA of the ZIKV was detectable by RT-qPCR at three days after infection in tree shrews, and it was undetectable after three days. Compared to RT-qPCR, the DAS-ELISA can monitor the presence of the NS1 protein from two to eight days. Therefore, the DAS-ELISA developed here can accurately and quickly detect the ZIKV.

Due to the lack of human clinical samples, we could not use the DAS-ELISA to screen the sera from patients infected with the ZIKV. In cases requiring the screening of clinical samples, the sera of ZIKV-infected patients were collected and added for DAS-ELISA according to the above procedures. Positive results were defined as the ratio of the value to the background value that was greater than 2.1, which was biologically significant. Furthermore, the positive results could be judged by the naked eye, as the solution was yellow after adding 50 µL of 2 M H_2_SO_4_. The DAS-ELISA developed here, like many other rapid detection methods, is not guaranteed to generate accurate diagnoses 100% of the time. Regardless of the positive results obtained from nucleic acid tests, the DAS-ELISA or another assays, still requires a second or even a third method to confirm the results.

In conclusion, a rapid, sensitive, and reliable method for the detection of ZIKV antigens was developed based on the sandwich ELISA. By targeting the NS1 protein, two Abs were prepared and screened by the DAS-ELISA. The specificity, sensitivity, and practical application of the DAS-ELISA were successfully established for monitoring ZIKV infection. Therefore, the DAS-ELISA developed in this work can serve as a potential tool for monitoring the virus in a variety of clinical samples and guiding the timely treatment of patients.

## Figures and Tables

**Figure 1 viruses-10-00634-f001:**
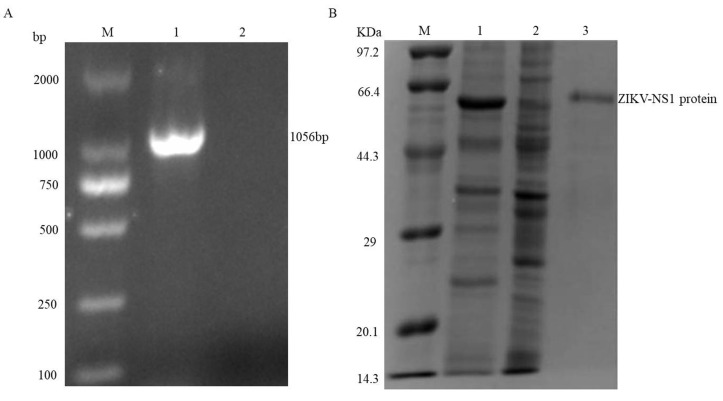
Analysis of the recombinant vector construction and expression of recombinant ZIKV-NS1 protein. (**A**) Identification of the positive clone of pET-32a-NS1 by bacteria polymerase chain reaction, lane 1: positive clone; lane 2: blank control. (**B**) Lane 1: *E. coli* was induced with IPTG; lane 2: *E. coli* before induced; lane 3: the purified ZIKV-NS1 after dialysis.

**Figure 2 viruses-10-00634-f002:**
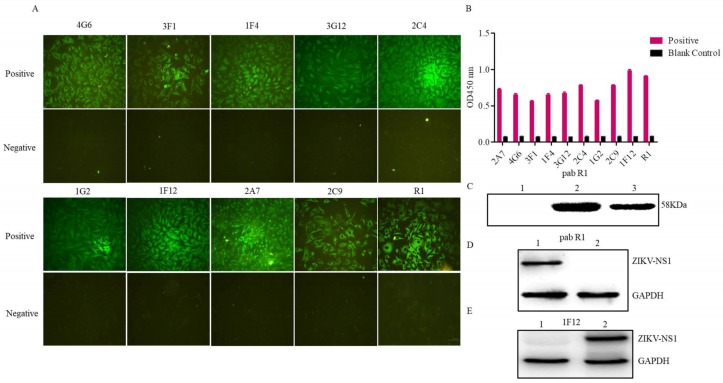
Evaluation of the pAb R1 and mAbs against ZIKV-NS1 protein. (**A**) Indirect immunofluorescent assay of the mAbs and pAb R1, , Scale bar = 100 μm. (**B**) ELISA test of the mAbs and pAb R1. (**C**) Western blot analysis of the pAb R1; lane 1: *E. coli* Rosetta (DE3) without IPTG; lane 2: *E. coli* Rosetta (DE3) induced with IPTG; lane 3: the purified ZIKV-NS1. (**D**) ZIKV infected (lane 1) and mock-infected Vero cells (lane 2) were used for Western blot analysis using pAb R1. (**E**) Mock-infected (lane 1) and ZIKV infected Vero cells (lane 2) were used for Western blot analysis using mAb 1F12.

**Figure 3 viruses-10-00634-f003:**
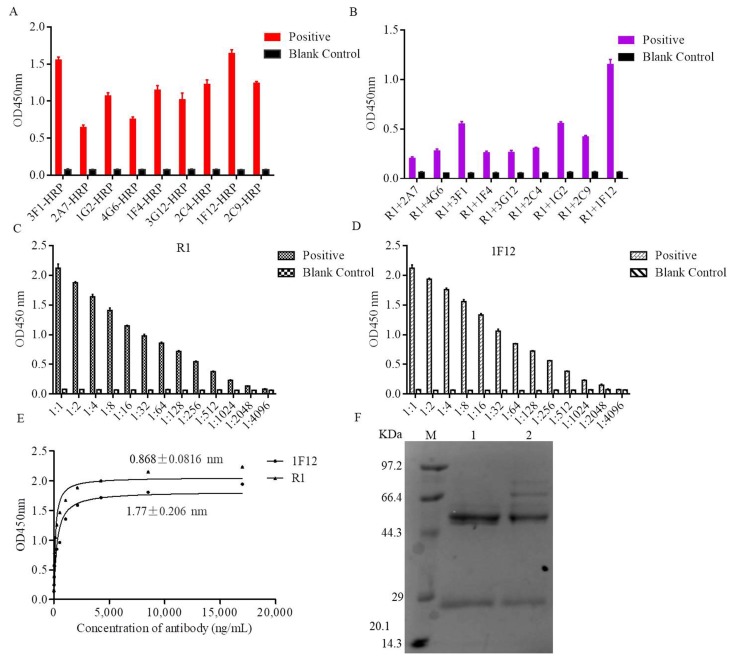
Characteristics of the key components of the DAS-ELISA. (**A**) The reactivity of nine mAbs conjugated with HRP, 5% skimmed milk as the blank control. (**B**) Nine different groups for DAS-ELISA, the blank control was 5% skimmed milk. The titer of pAb R1 (**C**) and mAb 1F12 (**D**) at different dilution ratios were determined by ELISA, 5% skimmed milk as the blank control. (**E**) The saturation curves for determination of the dissociation constants of mAb 1F12 and pAb R1. (**F**) SDS-PAGE analysis of purified mAb 1F12 (lane 1) and pAb R1 (lane 2).

**Figure 4 viruses-10-00634-f004:**
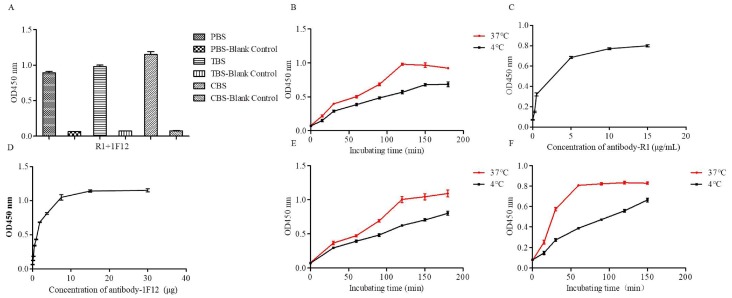
Optimization operations of DAS-ELISA. Optimum coating solution (**A**), coating temperatures and times (**B**) of pAb R1. Optimum working concentration of pAb R1 (**C**) and HRP-labeled mAb 1F12 (**D**). (**E**) Optimum incubating temperatures and times of antigen. (**F**) Optimum incubating times and temperatures of mAb 1F12-HRP probe.

**Figure 5 viruses-10-00634-f005:**
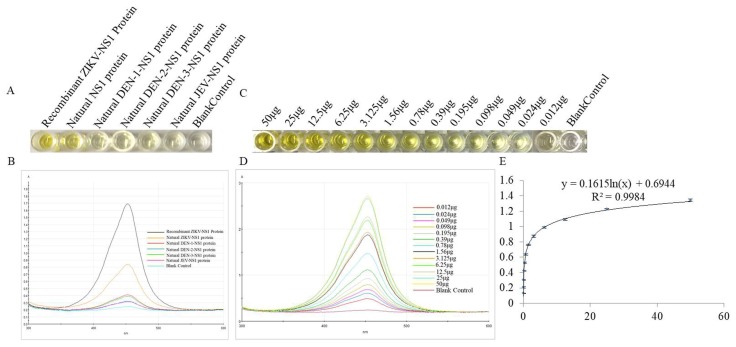
Specificity and sensitivity of the DAS-ELISA. (**A**) Specificity of the DAS-ELISA: well 1 represents the recombinant ZIKV-NS1 protein; well 2 represents the natural NS1 protein in the cell culture supernatant that infected with ZIKV; well 3−6 represent the natural NS1 protein in the cell culture supernatant infected with DEN-1, DEN-2, DEN-3, and JEV, respectively; and well 7 represents the blank control. (**B**) Optical density for the detection of different viruses. (**C**) Sensitivity of the DAS-ELISA: well 1−13 represent the different concentrations of recombinant ZIKV-NS1 proteins (500 µg mL^−1^–0.122 µg mL^−1^), well 14 represents the blank control. (**D**) Optical density for the detection of different concentrations of recombinant ZIKV-NS1 protein (500 µg mL^−1^–0.122 µg mL^−1^). (**E**) The plotted linear curve based on the different concentrations of recombinant ZIKV-NS1 protein.

**Figure 6 viruses-10-00634-f006:**
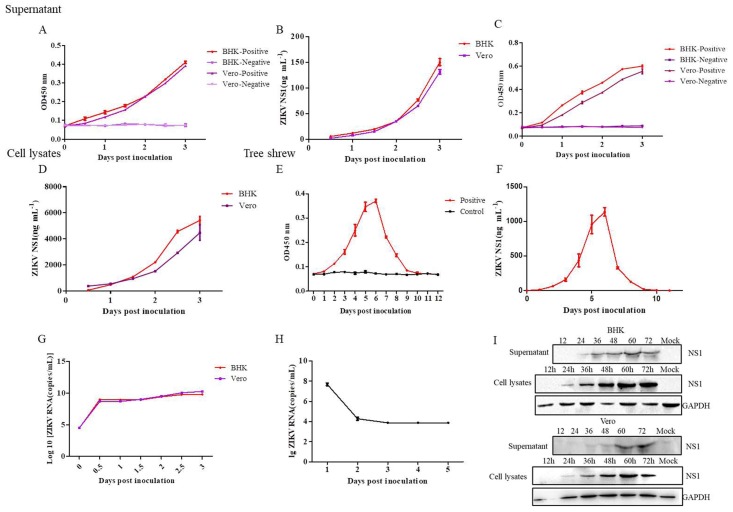
Detection of the ZIKV-NS1 protein in cell culture supernatants and lyates, as well as sera of animal infection model by the DAS-ELISA. Supernatants (**A**,**B**) and cell lysates (**C**,**D**) of Vero and BHK cells infected with ZIKV from 12, 24, 36, 48, 60, and 72 h were detected by DAS-ELISA. (**E**,**F**) Sera of tree shrews infected with ZIKV from 1 to 12 days were detected by DAS-ELISA. A Real-time quantitative reverse transcription polymerase chain reaction analysis of the supernatant of Vero, BHK (**G**) and the sera of tree shrews (**H**) after infected with ZIKV. (**I**) Western blot assay of supernatants and cell lysates of BHK and Vero cells infected with ZIKV.

**Table 1 viruses-10-00634-t001:** Primers used for RT-qPCR.

Name	Sequence	Length (bp)
ZIKV-ASF	GGTCAGCGTCCTCTCTAATAAACG	24
ZIKV-ASR	GCACCCTAGTGTCCACTTTTTCC	23
ZIKV-Probe	FAM-AGCCATGACCGACACCACACCGT-BQ1	23

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
