# Peer review of "Development and Characterization of Double-Antibody Sandwich ELISA for Detection of Zika Virus Infection"

_viruses, 2018, doi:10.3390/v10110634_

Round 1
Reviewer 1 Report
The work presented by Liding Zhang et al described an ELISA-based method to detect ZIKV NS1 protein with good specificity and sensitivity. Results were replicated in cell culture and animal model, showing the robust ability of this assay.
General comments:
· Generally improve grammar and sentence structure.
· Although it appears that the recombinant antigens can be accurately detected from living model systems, future experiments may benefit from expressing this recombinant antigen in eukaryotic cells to ensure any glycosylation patterns are maintained.
Specific comments:
· Line 72: please define the DAS-ELISA acronym.
· Lines 72-73: very redundant use of “detection”.
· Line 77: the accession number given is for a partial polyprotein sequence. Therefore, it is more accurate to either call it a protein sequence rather than a “genome”, or cite the accession number for the nucleotide sequence that was used for primer generation.
· Line 91: Recommend defining pab and mab for future use.
· Line 202: does “ascites” refer to antibodies in this sentence?
· Lines 209-220: Figures are listed in the text out of order (e.g. 3F before 3C, D).
· Figure 3: it is unclear whether “negative” samples in this experiment refers to no antibody added to the well, use of a negative control antibody, or some other control. Please clarify in the text and figure legend.
· Figure 4: legend is missing punctuation before “Optimum” on line 249.
· Figure 5: it is unclear from the text and figure legend whether well 2 (“ZIKV”) contains whole virus, recombinant protein, or natural NS1 protein. Please clarify.
· Line 280: detecting virus 8 days after exposure is excellent. However, future experiments will be needed to determine how this assay performs in human samples. More text about this point should be added in the Discussion section.
· Figure 6: it was unclear to me how the ELISA data was transformed to ng/mL in panel B.
· Figure 6: please change the order of the legend so that the description of each panel is in alphabetical order.
Additional thoughts:
· It was unclear to me whether a threshold was defined for differentiating positive results from negative results?
· Was any normalization process implemented on the ELISA data to facilitate cross-sample comparison?
· It would also be interesting to see how the performance of this platform compares to other commercially-available serology-based methods for detecting ZIKV proteins.
Author Response
Response to Reviewer 1 Comments
Point 1: Generally improve grammar and sentence structure.
Response 1: The grammar and sentence structure has been revised by Dolores Mruk.
Point 2: Line 72: please define the DAS-ELISA acronym.
Response 2: In line 74 of revised manuscript, we use “Double-antibody Sandwich ELISA” to define DAS-ELISA.
Point 3: Lines 72-73: very redundant use of “detection”.
Response 3: We use other words to replace the “detection” in lines 72-76 of revised manuscript.
Point 4: Line 77: the accession number given is for a partial polyprotein sequence. Therefore, it is more accurate to either call it a protein sequence rather than a “genome”, or cite the accession number for the nucleotide sequence that was used for primer generation.
Response 4: Thank you for your valuable advice, we use “The ZIKV-NS1 nucleotide sequence (Accession number: MG674719.1)” to replace the original sentence in revised manuscript.
Point 5: Line 91: Recommend defining pab and mab for future use.
Response 5: We redefine monoclonal antibody (mAb) and polyclonal antibody (pAb) for future use in line 94 of revised manuscript.
Point 6: Line 202: does “ascites” refer to antibodies in this sentence?
Response 6: The “ascites” in line 215 of revised manuscript were antibodies, which prepared by using anti-ZIKV-NS1 antibody positive hybridoma cells to inject female BALB/c mice. We have revised this sentence to make it much clearer.
Point 7: Lines 209-220: Figures are listed in the text out of order (e.g. 3F before 3C, D).
Response 7: In revised manuscript, we have been adjusted the Figures’ order.
Point 8: Figure 3: it is unclear whether “negative” samples in this experiment refers to no antibody added to the well, use of a negative control antibody, or some other control. Please clarify in the text and figure legend.
Response 8: In Figure 3, the “negative” samples were added 5% skimmed milk, which really means “Blank Control”. Thus, we use “Blank Control” to replace the “negative” in the revised manuscript.
Point 9: legend is missing punctuation before “Optimum” on line 249.
Response 9: In figure 4 of revised manuscript, we have been added the missing punctuation.
Point 10: Figure 5: it is unclear from the text and figure legend whether well 2 (“ZIKV”) contains whole virus, recombinant protein, or natural NS1 protein. Please clarify.
Response 10: In figure 5 of revised manuscript, the “ZIKV” means the “natural NS1 protein” in cell culture supernatant that infected with Zika virus. For, the natural NS1 protein can secrete in the cell culture supernatant with the soluble form during infection. We have revised the sentence of the figure legend to make it clear.
Point 11: Line 280: detecting virus 8 days after exposure is excellent. However, future experiments will be needed to determine how this assay performs in human samples. More text about this point should be added in the Discussion section.
Response 11: Thank you very much for your constructive suggestions. For we lack the human clinical samples, it is not clear whether the infection pattern of Zika virus in tree shrew animal models is consistent with humans, we revised the discussion for the DAS-ELISA assays performed in human samples in the Discussion section.
Point 12: it was unclear to me how the ELISA data was transformed to ng/mL in panel B.
Response 12: In figure 6, we use the figure 5E the plotted linear curve based on the different concentrations of recombinant ZIKV-NS1 protein to calculate the natural NS1 protein in supernatant, Cell lysates or the serum of Tree shrew model.
Point 13: Figure 6: please change the order of the legend so that the description of each panel is in alphabetical order.
Response 13: In figure 6 of the revised manuscript, we have been changed the order of the legend description of each panel in alphabetical order.
Point 14: It was unclear to me whether a threshold was defined for differentiating positive results from negative results?
Response 14: The positive results were defined as the ratio of positive value to background value that is greater than 2.1, which is of biological significance. And we have mentioned it in the discussion section of our revised manuscript.
Point 15: Was any normalization process implemented on the ELISA data to facilitate cross-sample comparison?
Response 15: Yes, all the ELISA chromogenic reactions are terminated at a fixed time, all data were subtracted from blank controls, and the each batch of experiments has corresponding standard curve for quantitative determination of antigen. The results of each group were calculated as the mean ± SD from three independent experiments.
Point 16: It would also be interesting to see how the performance of this platform compares to other commercially-available serology-based methods for detecting ZIKV proteins.
Response 16: This is a very useful suggestion. At the present stage the commercially-available serology-based methods for detection ZIKV are mainly focused on the ZIKV IgM/IgG detection kit, and a limited number of NS1 protein detection kits for ZIKV are still in the stage of trial. In the future, we will further screen antibody pairs to optimize the detection effect, and using other mature diagnostic kits to compare the detection results.
Reviewer 2 Report
The manuscript titled “Development and characterization of double-antibody sandwich ELISA for detection of Zika virus infection” by Zhang et al., details the development of a robust detection strategy for Zika virus. Focusing on detecting the NS1 protein as an early indicator of ZIKV infection, they characterize an ELISA-based approach to improve the detection of early infection. The creation of anti-NS1 antibodies is thoroughly detailed, and the specificity and affinity of the leading candidates is rigorously described.
All in all, the study is scientifically accurate and well controlled. The only concern scientifically is whether or not the individual samples assayed by the DS-ELISA were blinded to the researcher evaluating the utility of the assay. Regardless, there are no conceptual or technological deficits in the manuscript as presented.
The article is in need of significant editing for language and style.
Author Response
Response to Reviewer 2 Comments
Point 1: The article is in need of significant editing for language and style.
Response 1: Thank you for your valuable advice, the article has been significant edited for language and style by Dolores Mruk.
Point 2: The only concern scientifically is whether or not the individual samples assayed by the DS-ELISA were blinded to the researcher evaluating the utility of the assay.
Response 2: Thanks for your concern. ZIKV tree shrew model was built by our research group. Actually, in the DS-ELISA test for the samples, we used the different personnel of the authors to conduct the sample determination, respectively. Different samples are numbered only by numbers, especially for the animal model samples. Each sample we use different technologies measured, and these results were coincident. We think it accords with the double blind principle of scientific experiment.